# Difference in Perception of Onset of Old Age in Traditional (Hadza) and Modern (Polish) Societies

**DOI:** 10.3390/ijerph17197079

**Published:** 2020-09-27

**Authors:** Tomasz Frąckowiak, Agata Groyecka-Bernard, Anna Oleszkiewicz, Marina Butovskaya, Agnieszka Żelaźniewicz, Piotr Sorokowski

**Affiliations:** 1Institute of Psychology, University of Wroclaw, 50-527 Wroclaw, Poland; tomasz.frackowiak@uwr.edu.pl (T.F.); agata.groyecka@gmail.com (A.G.-B.); ania.oleszkiewicz@gmail.com (A.O.); 2Institute of Ethnology and Anthropology, Russian Academy of Sciences, Moscow 125993, Russia; marina.butovskaya@gmail.com; 3Taste and Smell Clinic, Technische Universität Dresden, 01307 Dresden, Germany; 4Russian State University for the Humanities, Moscow 125993, Russia; 5Department of Human Biology, University of Wroclaw, 50-137 Wroclaw, Poland; agnieszka.zelazniewicz@uwr.edu.pl

**Keywords:** aging perception, subjective age, older adults, elderly, senescence, cross-cultural studies, traditional population, Hadza

## Abstract

Despite relatively clear physiological indicators of old age, little is known about cross-cultural differences in psychological perceptions of the transition to old age. Although recent studies suggest consistency between modern countries, the subjective perception of old age onset in traditional societies remains poorly explored. Therefore, we compared the perception of timing of old age between a traditional tribe of hunter-gatherers (the Hadza) and a Polish sample representing a modern, industrialized population. The results indicate that the Hadza perceive old age onset as being significantly earlier than do the Poles. Furthermore, we found between-gender differences in the Polish sample: men set a lower threshold of old age onset than women. The Hadza showed no between-gender difference. Although the samples were matched for age, a larger proportion of Hadza considered themselves old. We discuss these findings from cultural and demographical perspectives.

## 1. Introduction

Determining the threshold of old age has become increasingly difficult owing to various demographic processes, such as prolongation of human life and population aging [1,2,3], which result from scientific advancements in public health, medicine, and technology. Such processes also include individual differences in both physical and psychological aging caused by genetics, lifetime health, life history, and lifestyle [4,5,6]. Most researchers identify the age of 65 years as the onset of old age [7,8] and this assumption is reflected in World Health Organization policies [9]. Beyond this physiological framework of old age, some empirical work has been conducted to understand the psychological perspective of aging (i.e., subjective individual perception of the timing of old age and inclusion of the self and others in the category “old people”).

Subjective perceptions of aging may reflect various changes, such as declining fitness and physical abilities, loss of health, new social roles (e.g., of a grandparent), death of a spouse and friends, retirement, need for organization of memories and life balance, dependency on others, and fear of death. Moreover, old age brings to mind changes in appearance, such as wrinkles, white hair, and baldness [10,11].

In addition to external criteria related to old age, some perceiver characteristics influence the perceived onset of old age. For example, as people become older, they tend to postpone their perceived transition to middle and old age [12]. Other factors that affect the perceived end of middle age (and therefore the onset of old age) are race (non-Whites perceive this transition as occurring earlier than Whites), self-rated health, and late parenthood [12].

One noteworthy construct that interacts with age perception is age identity. Subjective age describes how old or young a person feels [13]. Subjective age depends on several factors, including health and personality, and contributes to numerous psychological processes; for example, it predicts well-being, positive cognitive functioning, and longevity [14,15]. Interestingly, most people from their mid-20s to their 90s consider themselves younger than they actually are [16]. Some theories assume that this mechanism serves as self-protection against negative age-related stereotypes [17]. Positive interactions with grandchildren may also be associated with lower subjective age [15]. Subjective age determines whether individuals categorize themselves as old, and it shapes perceptions of the universal transition from one life stage to another [12].

Global stereotypes of older adults have several serious consequences [18]. Negative perceptions of aging may reduce the quality of life and longevity of older people [19,20,21,22]) and may even affect fluid cognitive functioning [23]. Exposure to stereotypical views about aging consistently impairs older adults’ performance, especially on cognitive and memory tasks [24]. This is consistent with the theory of stereotype embodiment [25], which suggests that the aging process and aging stereotypes are partly social constructs. Given the adverse consequences of negative stereotypes, it is crucial to identify who these stereotypes refer to.

Gender is an important factor that should be considered when studying perceptions of aging. There is a cross-species tendency for females to live longer than males, who typically present earlier senescence and lower life expectancies than females [26,27,28]. This sex difference is often explained by reproductive competition between males, which leads to their higher annual rates of mortality [29,30,31]. Men and women differ both as perceivers and as targets of age categorizations. These differences may be a result of evolutionary or sociocultural factors. Women have shorter periods of fertility [32] and have more severe biological costs of reproduction [33]. Relatedly, the perceptual boundaries of specific life periods in women may be more constrained and stage transitions may occur earlier [34,35,36]. Women’s social roles may also affect age categorizations, as family transition deadlines are broadly conceptualized as being culture dependent [34]. For example, middle and old age is thought to begin earlier in women than in men, which disadvantages women; in addition, the so-called double standards of aging mean that older women are evaluated more negatively than older men [37,38]. Both genders are devalued with age, but this shift is more pronounced for women, as their general evaluation depends to a greater extent on physical attractiveness (which declines with age); in contrast, men’s most valuable characteristic, earning potential, keeps growing [36]. Men’s judgments of the attractiveness of middle-aged women are harsher than women’s judgments [39], and men also tend to view middle age as both occurring and ending earlier than women do, especially when considering women’s lives. The same applies to evaluations of old age: men are perceived as reaching old age later than women, and the discrepancy in age norms is larger when evaluated by men [12,40]. Even women perceive themselves as old earlier than they perceive men [12,40], which is not surprising considering the pivotal role of physical condition and appearance in fulfilling gender cultural norms [41]. To explore the issue of gender, we included gender in this study and predicted that women would demonstrate later perceived onset of general old age than their men counterparts.

With some exceptions, e.g., [42], studies on the perception of the timing of old age have been limited to Western, industrialized samples. Beliefs about aging vary between cultures [43,44], but a recent study found consistent patterns of perception of the onset of old age across 26 modern and industrialized countries. Older adults worldwide were seen as less impulsive, less active, more agreeable, and more likely to stick to a routine than younger adults. Characteristics attributed to older people are consistent between self-reports and observational measures [42]. A particularly important aspect of this previous study [42] is the perceived timing of old age. The onset of old age as perceived by young people (in their early 20s) was relatively stable across the examined cultures (mean (M) = 59.6, standard deviation (SD) = 6.2; median (Mdn) = 58.3; Min = 47.6 (Malaysia); Max = 66.6 (Italy)).

Although the literature offers a perspective on the perceived timing of old age, little is known about such perceptions in traditional societies. Determining the perceived commencement of old age in traditional societies may elucidate the interaction between biological and social aging, as traditional societies have much less access to medical care or technical advancements than do modern societies. Traditional societies also live in relative isolation from other people and from Western influences and, therefore, also from media opinion and ideals that favor youthfulness [45,46]. These factors, together with evidence of positive stereotypes of old age [47] and greater happiness among older people in traditional societies compared with Europeans [22], make it difficult to predict differences in perceived onset of old age. However, they also suggest that it is unwise to assume that there are no differences in the perceptions of indigenous and Westernized people.

To address this gap in the literature on the perceived timing of the transition to old age among traditional societies, we performed a study of the Hadza, one of the last traditional hunter-gatherer societies of northern Africa [48,49]. Their population is estimated to be approximately 1200 people. This society has been extensively described in the literature [48,49,50,51]. They inhabit a territory around Lake Eyasi in northern Tanzania, living in remote camps located in the savannah–woodland highlands of the Mangola region. The Hadza are considered an egalitarian society with no clear hierarchy [22,50,52]. Importantly, regarding the gender issue discussed above, Hadza women have relatively high status compared with women in other societies [53]. This may be related to fewer negative stereotypes and less sexism, and may affect gender differences in perceived onset of old age. As hunting and gathering are the main activities of the Hadza, it is likely that they perceive individuals who cease performing these activities as old. Hadza men and women participate in these activities until they are unable to do so, but we are not aware of any studies that indicate when a Hadza typically stops contributing to provisioning. Therefore, there is no such thing as “retirement age” among the Hadza. Importantly, in behavior that reflects the “grandmother hypothesis” [54], Hadza grandmothers help their daughters to take care of their children [55], which contributes to community cohesion. However, Hadza women have grandchildren relatively early compared with Western women, which may be of importance in assessing perceived onset of old age.

We compared the Hadza results with those from a modern, industrialized population from Poland. As part of a Western society, the Polish sample had a longer life expectancy owing to access to medical and social care, technological advancements, higher socioeconomic status, and relative independence from ecological factors, such as extreme weather conditions or hunger. The main study aim was to explore the perceived onset of old age among Polish and Hadza participants, taking into consideration sex differences.

## 2. Materials and Methods

### 2.1. Participants

We recruited 96 Hadza (47 women, M_age_ = 37.8; SD_age_ = 15.1; Mdn_age_ = 36.5, Min_age_ = 17.0; Max_age_ = 75.0) and 124 Poles, approximately matched for gender and age (66 women, M_age_ = 36.0; SD_age_ = 13.4; Mdn_age_ = 35.0; Min_age_ = 18.0; Max_age_ = 76.0). Data from three Hadza were removed from the analysis because the individuals declined to answer the question about the threshold of old age. Two individuals justified their decision by stating that “old age is when one has grandchildren.” There was no significant difference in gender distribution, χ2(1) = 0.39, *p* = 0.53 or age, t(218) = 0.93, *p* = 0.35, between the two populations.

The Hadza were recruited in the Mangola region, close to the camps they inhabit. Polish participants were recruited in Brzeg (a small provincial city in Poland) during educational courses of pedagogy. Subjects participated in the study voluntarily during a break in the course.

### 2.2. Ethics

Participants provided informed consent before being included in the study and were notified that they could quit the study at any time. As almost all the Hadza are illiterate, participants gave their informed oral consent. The complete study protocol and both consent procedures were approved by the Commission for Science and Technology of Tanzania and by the Institutional Review Board of the University of Wroclaw (Wroclaw, Poland). The study was conducted in accordance with the Declaration of Helsinki.

### 2.3. Procedure

The same procedure was used for the Polish and Hadza samples. Participants were interviewed orally during individual sessions. During interviews with Hadza participants, the researcher was accompanied by a local assistant who was fluent in Swahili and familiar with the Hadza language. Participants were assured that they could quit the interview at any time with no adverse consequences.

Hadza participants were also interviewed to obtain data for other studies (unrelated to this one) and each full interview took approximately 15 min per participant. However, responding to the present study questions took less than 5 min for both the Hadza and Polish participants. There were no additional studies concurrent with the present one in Poland.

### 2.4. Measures

Participants were first asked to indicate three persons they knew who they considered old:
“Please, list three people that you consider old.”

The beginning of old age was set as the age of the youngest of the old people mentioned by the participant. Some Hadza participants were unaware of their exact age. They provided an approximate estimation of how old they and the people they listed were. These ages were subsequently confirmed by one author (M.B.), who has been conducting research among the Hadza for over 10 years and knew our participants and the people surrounding them. In this stage, we identified participants’ ages and the ages of the three old people each participant mentioned. The lowest of these three ages (for the three persons listed by each participant) was treated as a dependent variable. Then we asked participants whether they considered themselves old:
“Do you consider yourself old?”

Possible responses were no (coded as 0) and yes (coded as 1), and were considered an index of self-categorization as an old person.

The same method was used in Poland, except that the Polish participants were able to confirm the age of the three people they labeled as old.

### 2.5. Hypotheses and Statistical Analyses

We expected that the Hadza participants would demonstrate an earlier perceived onset of old age than the Polish participants, and that women of both populations would demonstrate a later perceived onset than men. We also controlled for participant age, as the perceived transition to old age may shift with age.

Models were estimated using IBM SPSS software v. 24, with the level of significance set to α = 0.05. We first performed a univariate analysis of variance with perceived onset of old age as the dependent variable, group (Hadza vs. Poles) and gender (women vs. men) as factors, and age as a covariate. We examined the main effects of group and gender and their interaction. Furthermore, we analyzed the independence of self-categorization as an old person by group (Hadza vs. Poles) and gender (women vs. men) using the chi-squared test. The significance level was Bonferroni corrected for a double comparison.

## 3. Results

Univariate analysis of variance revealed a significant interaction between group and gender, F(1, 212) = 6.02, *p* = 0.015, η2 = 0.02, indicating that the perceived onset of old age in the Hadza group was significantly earlier (M = 54 ± 0.8 years) than in the Polish group (M = 68.5 ± 0.7 years). Post hoc comparisons confirmed that this difference was significant in both men and women (*p*s < 0.001). There was no significant between-gender difference in Hadza participants, but Polish men had a lower perceived threshold of old age onset (M = 66.3 ± 1 years) than Polish women (M = 70.6 ± 0.9 years; *p* = 0.002; Figure 1). The effect of participant age on the perceived old age threshold was not significant (*p* = 0.09).

Earlier perceived onset of old age was also reflected in self-categorizations made by the Hadza and Polish samples. Although the samples were of similar age, significantly more Hadza considered themselves old (29.2%) than did Poles (12.9%), χ^2^ (1) = 8.9, *p* = 0.003 (see Figure 2). Self-categorization as old was independent of participant gender (*p* = 0.89).

## 4. Discussion

The present study, for the first time, compared perceptions of the timing of aging in the traditional Hadza tribe with a gender- and age-matched sample of Poles, representing a modern society. The results of our study further confirm that perceptions of aging vary between cultures [22,43,44,56] and extend previous findings by providing a perspective on the perception of old age onset in a traditional population. We observed that persons categorized by the Hadza as old were significantly younger than those considered old by the Polish sample. Furthermore, individuals categorized as old by Polish women were significantly older than those considered old by Polish men.

Level of economic development is related to life expectancy [57,58,59]. It seems intuitive that in societies with longer life expectancy, individual experience and observation would shape beliefs about aging. A person surrounded by many older people as they grow up may have more salient exemplars of old people to name than a person surrounded by fewer older people or living in a population with a shorter life expectancy. Like other indigenous tribes, the average Hadza life expectancy at birth is low (32.5 years) [60,61] compared with Poland (78.2 years) [62]. However, this lower average life expectancy does not necessarily mean that the Hadza live very short lives. Rather, low life expectancy is related to high mortality in the first few years of life, rather than a short life in general [49,61]. For example, a Hadza woman who reaches the age of 45 years still has more than 20 years of life expectancy [25]. In other words, although there is substantial variation across human groups in life expectancy at early ages, there is substantial convergence after approximately 30 years of age [61].

Therefore, the Hadza may have few examples of old people in their society, but these prototypes may not necessarily be younger than the examples available to Poles. Comparing the Hadza data with results reported by Chan and colleagues [42] shows that there are some industrialized societies in which people are considered to reach old age even earlier (e.g., Malaysia; 47.6 years); in contrast, the Hadza believe that people become old at the age of approximately 54 years. Therefore, although the perceived transition to old age in hunter-gatherers appears to be earlier than for Western populations, this difference is not extreme, which is also indicated by the small effect size. The Hadza perception of old age onset may not be solely related to their very low exposure to industrialized lifestyles and the available prototypes of old people, but may also reflect a deep attachment to their own culture and lifestyle.

This field study revealed an additional insight about Hadza perceptions of the boundaries of old age. Although participants were not explicitly asked to indicate attributes or life events associated with old age, some spontaneously shared this information with us. The most common comments they made during the interview were that someone turns old “once he/she has grandchildren” or “once he or she has white hair.” One participant claimed that “one is old when he or she has an adult child.” This suggests that although they have difficulty defining their own or another person’s age, the Hadza have a definite picture of old age that is convergent with Western concepts. It is therefore not surprising that they are more likely to categorize a younger person as old, as transitions to parenthood, grandparenthood, or becoming an active helper in feeding the family occur much earlier among the Hadza [49,63].

We found that Polish participants had a significantly higher threshold of perceived old age than Hadza participants. This is in line with many previous findings on the positive relationship between the level of industrialization and life expectancy [57,58,59,64]. Interestingly, we found a significant between-gender difference in perceptions of aging: Polish men categorized significantly younger individuals as old compared to Polish women. This may reflect previously reported Western cultural ideals of beauty (these particularly affect women, who are subjected to a stronger pressure to strive for beauty and remain young looking, especially in men’s eyes) and different expectations of aging between men and women [37,65,66]. Moreover, the lack of sex differences in the Hadza data may be attributed to the highly egalitarian nature of their culture and the high status of women. Previous studies have also reported a lack of sex differences in the Hadza for phenomena which show highly pronounced sex differences in other cultures, e.g., [67].

One limitation of this study is the imperfect measure of the onset of old age. Participants were asked to select three people they perceived as old. We cannot be sure whether there were other, younger, individuals who they also perceived as old. However, for the Hadza participants particularly, it seemed more appropriate to ask indirectly about the onset of old age rather than using more direct questions. We kept the procedure as similar as possible for the Polish participants. Additionally, although this study provides interesting insights into perceptions of old age in a traditional society, we did not identify any specific mechanism behind this effect. Additional studies are needed to replicate and explain the differences found here. Finally, future studies should take into account participant health status, which may affect perceptions of elderly people.

## 5. Conclusions

Our results provide evidence that members of an indigenous society, who lead lives similar to those of ancestral humans, perceive old age as beginning significantly earlier than do Polish participants of a modern, industrialized European society. According to the Hadza, old age starts relatively early, but not extremely early. One important difference between the two societies is that there was a gender difference in Poles but not in the Hadza. This suggests that women and men have different perceptions of old age and that this difference may have developed at some later point in the evolutionary history of humans.

## Figures and Tables

**Figure 1 ijerph-17-07079-f001:**
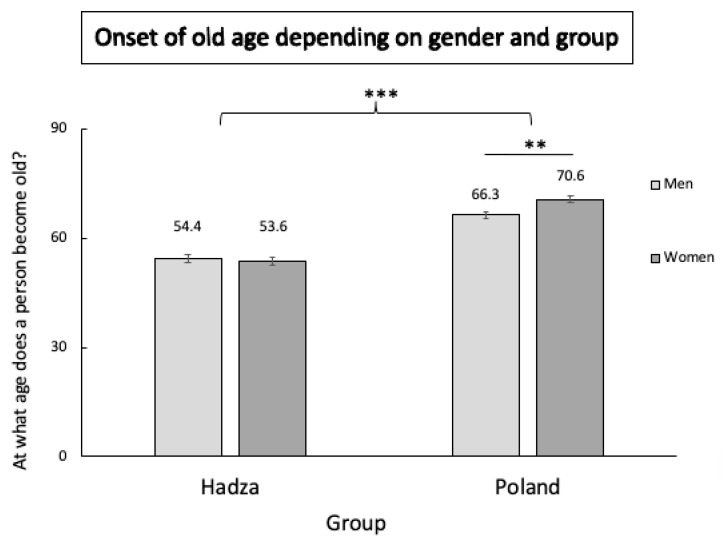
Distribution of perceived onset of old age in Polish (N_Women_ = 66 and N_Men_ = 58) and Hadza (N_Women_ = 47 and N_Men_ = 49) men and women. ** *p* < 0.01, *** *p* < 0.001.

**Figure 2 ijerph-17-07079-f002:**
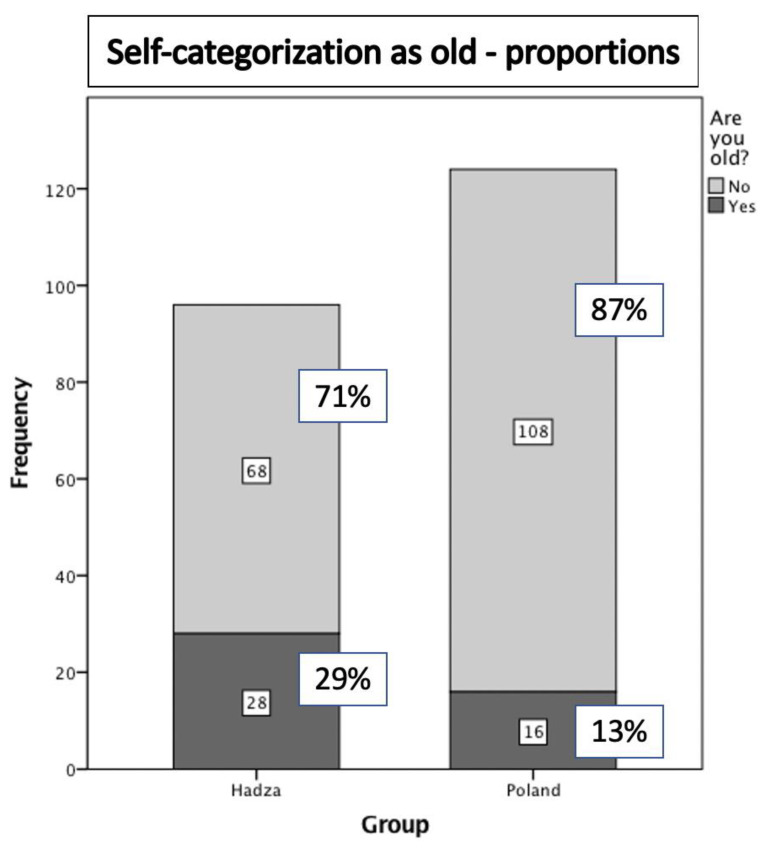
Distribution of self-categorization as old in Polish and Hadza samples.

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
