# Peer review of "Difference in Perception of Onset of Old Age in Traditional (Hadza) and Modern (Polish) Societies"

_ijerph, 2020, doi:10.3390/ijerph17197079_

Round 1

Reviewer 1 Report

This study’s focus on Hazda people is really noteworthy.  As the authors point out, most studies of perceptions of life course timing have focused on Western populations. However, the manuscript could be strengthened in several regards.

  1. The prior studies on perceived timing of later life stages could be reviewed more thoroughly and used to shape this study’s predictions, as well as interpret its findings. For example, prior studies find that people tend to postpone the timing of middle age and old age as they themselves approach these ages.  In other words, younger people tend to view these life stages as occurring earlier than do people who are nearing or in these life stages.  The literature also reports that sometimes people think of social roles (e.g., being a grandparent or not working for pay) or experiencing health declines as markers of old age.  These findings should be more thoroughly reviewed in the front of the paper, where it would also be helpful to find discussions of how these patterns may or may not be expected to exist in traditional societies.  Outlining this context carefully in the front of the paper would clarify your study’s contribution.

  1. Related to #1, prior studies also have examined how gender shapes constructions of life stages – but this work isn’t covered in the front of the paper. I’d also note that the discussion of gender is problematic in its reliance on demographic and biological explanations.  Gender is a social construct that shapes experiences and constructions of aging.  Related to your study, the onset of various life stages, like middle age and old age tend to be constructed as occurring earlier for women than men. Further, some work finds that men view these stages as occurring earlier than do women – and they view these stages as occurring especially early for women (e.g., Toothman & Barrett, 2012).  But the explanation doesn’t lie in biology or demography – a conclusion that is, in fact, consistent with note in the introduction regarding women’s longer life expectancy (which would seem to lead to a postponement of the onset of later life stages).  In short, more attention should be given to the arguments in the paper that relate to gender. The work of feminist gerontologists (e.g, Barrett, Calasanti, Hurd Clarke) will be helpful in this regard.

  1. More details on the study are needed. How long did the interviews last, on average? How old were the Polish participants?  Because they were students, I suspect they are younger than the Hazda participants, which makes it more difficult to compare their perceptions of age.  It would be helpful to comment on the implications of such differences (or others).

  1. The manuscript should be edited for English writing conventions. For example, the sentence in the abstract that includes “however” is actually a run-on sentence (aka comma splice).  This issue could be remedied by putting a semicolon before “however.”

Author Response

11.09.2020

Dear Editor,

My co-authors and I would like to thank you for your time devoted to evaluation of our manuscript and for the possibility to revise it. We are also grateful to two independent reviewers for their helpful feedback. We carefully considered all comments and revised the manuscript accordingly. Major changes have been made in the introduction, following reviewers’ recommendations. The manuscript has also been proofread by professional editing company. The changes are identified by blue text in the manuscript and are addressed point-by-point below. In the responses, we refer to line numbers corresponding with the clear version of the manuscript (with accepted changes).

With kind regards,

Piotr Sorokowski and co-authors

Reviewer 1

This study’s focus on Hazda people is really noteworthy.  As the authors point out, most studies of perceptions of life course timing have focused on Western populations. However, the manuscript could be strengthened in several regards.

We are glad that the Reviewer in general appreciated our work and contribution. We thank for Reviewer’s thoughtful remarks and we believe that the manuscript benefited from them substantially.

  1. The prior studies on perceived timing of later life stages could be reviewed more thoroughly and used to shape this study’s predictions, as well as interpret its findings. For example, prior studies find that people tend to postpone the timing of middle age and old age as they themselves approach these ages.  In other words, younger people tend to view these life stages as occurring earlier than do people who are nearing or in these life stages.  The literature also reports that sometimes people think of social roles (e.g., being a grandparent or not working for pay) or experiencing health declines as markers of old age.  These findings should be more thoroughly reviewed in the front of the paper, where it would also be helpful to find discussions of how these patterns may or may not be expected to exist in traditional societies.  Outlining this context carefully in the front of the paper would clarify your study’s contribution.

Thank you for this comment. All abovementioned issues have been raised in the revised version of the introduction.

We considered the role of one’s own age in perception of onset of old age (lines 43-44), as well as of other factors.

We also looked closer at the issues of:

-social roles in old age, both in western- and Hadza societies   

“Subjective perceptions of aging may reflect various changes such as declining fitness and physical abilities, loss of health, new social roles (e.g., of a grandparent), death of a spouse and friends, retirement, need for organization of memories and life balance, dependency on others, and fear of death. Moreover, old age brings to mind changes in appearance, such as wrinkles, white hair, and baldness [10,11].” (lines 37-41)

-potential characteristics of the perceiver that contribute to the perception of timing in old age

“In addition to external criteria related to old age, some perceiver characteristics influence the perceived onset of old age. For example, as people become older, they tend to postpone their perceived transition to middle and old age [12]. Other factors that affect the perceived end of middle age (and therefore the onset of old age) are race (Non-Whites perceive this transition as occurring earlier than Whites), self-rated health, and late parenthood [12].” (lines 42-46)

-subjective age

“One noteworthy construct that interacts with age perception is age identity. Subjective age describes how old or young a person feels [13]. Subjective age depends on several factors, including health and personality, and contributes to numerous psychological processes; for example, it predicts well-being, positive cognitive functioning, and longevity [14,15]. Interestingly, most people from their mid 20s to their 90s consider themselves younger than they actually are [16]. Some theories assume that this mechanism serves as self-protection against negative age-related stereotypes [17]. Positive interactions with grandchildren may also be associated with lower subjective age [15]. Subjective age determines whether individuals categorize themselves as old, and it shapes perceptions of the universal transition from one life stage to another [12].” (lines 47-55)

The issue of social roles have also been used to discuss our results:

“It is therefore not surprising that they are more likely to categorize a younger person as old, as transitions to parenthood, grandparenthood, or becoming an active helper in feeding the family occur much earlier among the Hadza [49,63].” (lines 252-254)

  1. Related to #1, prior studies also have examined how gender shapes constructions of life stages – but this work isn’t covered in the front of the paper. I’d also note that the discussion of gender is problematic in its reliance on demographic and biological explanations.  Gender is a social construct that shapes experiences and constructions of aging.  Related to your study, the onset of various life stages, like middle age and old age tend to be constructed as occurring earlier for women than men. Further, some work finds that men view these stages as occurring earlier than do women – and they view these stages as occurring especially early for women (e.g., Toothman & Barrett, 2012).  But the explanation doesn’t lie in biology or demography – a conclusion that is, in fact, consistent with note in the introduction regarding women’s longer life expectancy (which would seem to lead to a postponement of the onset of later life stages).  In short, more attention should be given to the arguments in the paper that relate to gender. The work of feminist gerontologists (e.g, Barrett, Calasanti, Hurd Clarke) will be helpful in this regard.

Thank you for this comment. We added a broad paragraph regarding the gender differences in construction of live stages and we fully agree that biological or demographical explanations are insufficient.

“Gender is an important factor that should be considered when studying perceptions of aging. There is a cross-species tendency for females to live longer than males, who typically present earlier senescence and lower life expectancies than females [26–28]. This sex difference is often explained by reproductive competition between males, which leads to their higher annual rates of mortality [29–31]. Men and women differ both as perceivers and as targets of age categorizations. These differences may be a result of evolutionary or sociocultural factors. Women have shorter periods of fertility [32] and have more severe biological costs of reproduction [33]. Relatedly, the perceptual boundaries of specific life periods in women may be more constrained and stage transitions may occur earlier [34–36]. Women’s social roles may also affect age categorizations, as family transition deadlines are broadly conceptualized as culture dependent [34]. For example, middle and old age is thought to begin earlier in women than in men, which disadvantages women; in addition, the so-called double standards of aging mean that older women are evaluated more negatively than older men [37,38]. Both genders are devalued with age, but this shift is more pronounced for women, as their general evaluation depends to a greater extent on physical attractiveness (which declines with age); in contrast, men’s most valuable characteristic, earning potential, keeps growing [36]. Men’s judgments of the attractiveness of middle-aged women are harsher than women’s judgments [39], and men also tend to view middle age as both occurring and ending earlier than women do, especially when considering women’s lives. The same applies to evaluations of old age: men are perceived as reaching old age later than women, and the discrepancy in age norms is larger when evaluated by men [12,40]. Even women perceive themselves as old earlier than they perceive men [12,40], which is not surprising considering the pivotal role of physical condition and appearance in fulfilling gender cultural norms [41]. To explore the issue of gender, we included gender in this study and predicted that women would demonstrate later perceived onset of general old age than their men counterparts.” (lines 63-86)

  1. More details on the study are needed. How long did the interviews last, on average? How old were the Polish participants?  Because they were students, I suspect they are younger than the Hazda participants, which makes it more difficult to compare their perceptions of age.  It would be helpful to comment on the implications of such differences (or others).

Our Polish sample was not comprised of students. We recruited attendees of educational courses and they were matched in terms of age to the Hadza sample:

“We recruited 96 Hadza (47 women, Mage = 37.8; SDage = 15.1; Mdnage = 36.5, Minage = 17.0; Maxage = 75.0) and 124 Poles, approximately matched for gender and age (66 women, Mage = 36.0; SDage = 13.4; Mdnage = 35.0; Minage = 18.0; Maxage = 76.0).(lines 136-138)

“There was no significant difference in gender distribution, χ2(1) = 0.39, p = 0.53 or age, t(218) = 0.93, p = 0.35 between the two populations.” (lines 140-142)

We also added some details on the study:

“Hadza participants were also interviewed to obtain data for other studies (unrelated to this one) and each full interview took approximately 15 minutes per participant. However, responding to the present study questions took less than 5 minutes for both the Hadza and Polish participants. There were no additional studies concurrent with the present one in Poland.” (lines 160-163)

  1. The manuscript should be edited for English writing conventions. For example, the sentence in the abstract that includes “however” is actually a run-on sentence (aka comma splice).  This issue could be remedied by putting a semicolon before “however.”

The manuscript has been proofread by a professional editing company (Edanz).

Reviewer 2 Report

Major points that need clarification:

1) The title is vague and unclear and not a reflection of the study.  I suggest something along the lines of: "Difference in perception of onset of old age..."

2) Throughout the document the phrase "an old age" is used.  It would be more scientific to perhaps refer to aging (line 13, 16) or leaving "an" (line 17) out? Line 19, 21, 22, 28, 33, etc. Perhaps it is best to consult with a language expert on this.

3) Line 49 refers to "consequences of stereotypes..." needs to be unpacked and elaborated on.

 4) Line 67 & 68 refer to "native to human kind" - This statement is unclear as well as line 69 referring to "definitional inaccuracies" need further explanation.  

5) I am of the opinion that the gap in literature (line 70) has not clearly been explained. 

6) The hypothesis in line 77 is out of place and should be moved to "Methodologies".

7) Line 91 - 95 belongs in the Introduction or Lit Review section and not under "Participants".

8) In the "Methodology" section, it is not what instruments were used or what questions were asked. The reference to ethics (line 97 and 101, is insufficient and at least a paragraph should reflect on the ethical conduct of the researchers or study. These references to "Informed consent" should be moved to the past paragraphs before "Procedure". Line 114 and 115 should also move to the Ethics paragraph.

9) The graphs on p.4 have no title, only a description.

10) Line 158 - 159 is unclear and needs to be elaborated on.

11) The researchers refer to a gap in literature in line 68, but in the "Discussion" section declare that the study "extends" the literature. There are two distinct differences.

Minor points:

1) Line 42 - "counties", should this not be "countries"?

2) Line 67 & 68 refer to "native to humand kind" - This statement is unclear as well as line 69?

3) Line 129 contains a minor typing error.

Author Response

11.09.2020

Dear Editor,

My co-authors and I would like to thank you for your time devoted to evaluation of our manuscript and for the possibility to revise it. We are also grateful to two independent reviewers for their helpful feedback. We carefully considered all comments and revised the manuscript accordingly. Major changes have been made in the introduction, following reviewers’ recommendations. The manuscript has also been proofread by professional editing company. The changes are identified by blue text in the manuscript and are addressed point-by-point below. In the responses, we refer to line numbers corresponding with the clear version of the manuscript (with accepted changes).

With kind regards,

Piotr Sorokowski and co-authors

Reviewer 2

Major points that need clarification:

  • The title is vague and unclear and not a reflection of the study.  I suggest something along the lines of: "Difference in perception of onset of old age..."

Thank you for this remark, we followed your suggestion. The new title reads:

“Difference in perception of onset of old age in traditional (Hadza) and modern (Polish) societies.”

  • Throughout the document the phrase "an old age" is used.  It would be more scientific to perhaps refer to aging (line 13, 16) or leaving "an" (line 17) out? Line 19, 21, 22, 28, 33, etc. Perhaps it is best to consult with a language expert on this.

The manuscript has been carefully proofread by a language expert (Edanz). The phrase “an old age” does not appear in the manuscript anymore.

  • Line 49 refers to "consequences of stereotypes..." needs to be unpacked and elaborated on.

We have expanded this notion on consequences of stereotypes and the revised paragraphs read:

“Global stereotypes of older adults have several serious consequences [18]. Negative perceptions of aging may reduce the quality of life and longevity of older people [19–22]) and may even affect fluid cognitive functioning [23]. Exposure to stereotypical views about aging consistently impairs older adults’ performance, especially on cognitive and memory tasks [24]. This is consistent with the theory of stereotype embodiment [25], which suggests that the aging process and aging stereotypes are partly social constructs. Given the adverse consequences of negative stereotypes, it is crucial to identify who these stereotypes refer to.” (lines 56-62)

  • Line 67 & 68 refer to "native to human kind" - This statement is unclear as well as line 69 referring to "definitional inaccuracies" need further explanation.  

We agree that this statement was inaccurate and, in fact, the entire paragraph was too speculative. Therefore, we decided to erase it.  

  • I am of the opinion that the gap in literature (line 70) has not clearly been explained. 

We expanded the paragraph above this statement explaining more thoroughly why examining perception of traditional societies is important.

“Determining the perceived commencement of old age in traditional societies may elucidate the interaction between biological and social aging, as traditional societies have much less access to medical care or technical advancements than do modern societies. Traditional societies also live in relative isolation from other people and from Western influences and, therefore, also from media opinion and ideals that favor youthfulness [45,46]. These factors, together with evidence of positive stereotypes of old age [47] and greater happiness among older people in traditional societies compared with Europeans [22], make it difficult to predict differences in perceived onset of old age. However, they also suggest that it is unwise to assume that there are no differences in the perceptions of indigenous and Westernized people.” (lines 98-107)

Moreover, we highlighted why studying the perception of life stages is useful in general (we referred to it in the response to comment 3).

6) The hypothesis in line 77 is out of place and should be moved to "Methodologies".

Thank you for this comment. We added hypothesis to the Method section.

7) Line 91 - 95 belongs in the Introduction or Lit Review section and not under "Participants".

We shifted these lines up to the introduction.

8) In the "Methodology" section, it is not what instruments were used or what questions were asked. The reference to ethics (line 97 and 101, is insufficient and at least a paragraph should reflect on the ethical conduct of the researchers or study. These references to "Informed consent" should be moved to the past paragraphs before "Procedure". Line 114 and 115 should also move to the Ethics paragraph.

We created separate ethic section. (lines 147-153)

We admit that Procedure and Measures subsections were combined together which

might have been disturbing. In order to facilitate perception of our paper, we separated

these sections and more literally referred to the questions that we used. The revised parts read:

“2.2. Procedure

The same procedure was used for the Polish and Hadza samples. Participants were interviewed orally during individual sessions. During interviews with Hadza participants, the researcher was accompanied by a local assistant who was fluent in Swahili and familiar with the Hadza language. Participants were assured that they could quit the interview at any time with no adverse consequences. 

Hadza participants were also interviewed to obtain data for other studies (unrelated to this one) and each full interview took approximately 15 minutes per participant. However, responding to the present study questions took less than 5 minutes for both the Hadza and Polish participants. There were no additional studies concurrent with the present one in Poland.

2..3 Measures

Participants were first asked to indicate three persons they knew who they considered old:

“Please, list three people that you consider old.”

The beginning of old age was set as the age of the youngest of the old people mentioned by the participant. Some Hadza participants were unaware of their exact age. They provided an approximate estimation of how old they and the people they listed were. These ages were subsequently confirmed by one author (M.B.), who has been conducting research among the Hadza for over 10 years and knew our participants and the people surrounding them. In this stage, we identified participants’ ages and the ages of the three old people each participant mentioned. The lowest of these three ages (for the three persons listed by each participant) was treated as a dependent variable. Then we asked participants whether they considered themselves old: 

“Do you consider yourself old?”

Possible responses were no (coded as 0) and yes (coded as 1), and were considered an index of self-categorization as an old person.

The same method was used in Poland, except that the Polish participants were able to confirm the age of the three people they labeled as old.”

9) The graphs on p.4 have no title, only a description.

We added titles to both figures.

10) Line 158 - 159 is unclear and needs to be elaborated on.

We followed your suggestion and formulated our thought in other words too.:

„In other words, although there is substantial variation across human groups in life expectancy at early ages, there is substantial convergence after approximately age 30 years [61].” (lines 232-234)

11) The researchers refer to a gap in literature in line 68, but in the "Discussion" section declare that the study "extends" the literature. There are two distinct differences.

We changed the wording to make it consistent. In short, what we mean is that this study tests something that has not been tested before and that the research question that we state is theoretically justified. 

Minor points:

  • Line 42 - "counties", should this not be "countries"?

Thank you.

2) Line 67 & 68 refer to "native to humand kind" - This statement is unclear as well as line 69?

As noted earlier, this paragraph has been deleted.

3) Line 129 contains a minor typing error.

All typing errors have been corrected by Diane Williams, representing a professional editing company.

Round 2

Reviewer 2 Report

The study investigates the perception of the onset of old age amongst modern (Polish) and otherwise traditional (Hazda) populations. The manuscript claims that traditional societies perceive the onset of old age earlier than modern societies. The sample size is sufficient and the methodologies are clearly outlined with results discussed from cultural and demographical perspectives.

The authors have addressed many issues and made changes to the document in that it reads easier and arguments flow better. The title is improved and the study now is clear with regards to the research question in the form of a single question asked from the participants.